# DNA methylation profiling to determine the primary sites of metastatic cancers using formalin-fixed paraffin-embedded tissues

Shirong Zhang[1,2,15], Shutao He[3,4,15], Xin Zhu[5,15], Yunfei Wang[6], Qionghuan Xie[6], Xianrang Song[7], Chunwei Xu[8], Wenxian Wang[5], Ligang Xing[7], Chengqing Xia[6], Qian Wang[9], Wenfeng Li[10], Xiaochen Zhang[11], Jinming Yu[7], Shenglin Ma[1,12] ✉, Jiantao Shi[3] ✉ & Hongcang Gu[13,14] ✉

Identifying the primary site of metastatic cancer is critical to guiding the subsequent treatment. Approximately 3–9% of metastatic patients are diagnosed with cancer of unknown primary sites (CUP) even after a comprehensive diagnostic workup. However, a widely accepted molecular test is still not available. Here, we report a method that applies formalin-fixed, paraffin-embedded tissues to construct reduced representation bisulfite sequencing libraries (FFPE-RRBS). We then generate and systematically evaluate 28 molecular classifiers, built on four DNA methylation scoring methods and seven machine learning approaches, using the RRBS library dataset of 498 fresh-frozen tumor tissues from primary cancer patients. Among these classifiers, the **be**ta value-based **li**near support **ve**ctor (BELIVE) performs the best, achieving overall accuracies of 81-93% for identifying the primary sites in 215 metastatic patients using top-k predictions (k = 1, 2, 3). Coincidentally, BELIVE also successfully predicts the tissue of origin in 81-93% of CUP patients (n = 68).

Cancer causes approximately 10 million deaths worldwide and more than 3 million in China alone[1,2], of which 90% are due to metastasis[3]. For most patients with metastatic cancer, the tissue of origin can be determined by a comprehensive diagnostic workup, either in the early or metastatic stages[4]. However, the remaining 3–9% of cases have to be assigned as cancer of unknown primary sites, making this heterogeneous group of cancers the seventh or eighth most frequent malignancy and the fourth most common cause of cancer death[5–8].

[1]Translational Medicine Research Center, Hangzhou First People's Hospital, 310006 Hangzhou, Zhejiang Province, China. [2]Key Laboratory of Clinical Cancer Pharmacology and Toxicology Research of Zhejiang Province, Hangzhou First People's Hospital, 310006 Hangzhou, Zhejiang Province, China. [3]State Key Laboratory of Molecular Biology, Shanghai Institute of Biochemistry and Cell Biology, Center for Excellence in Molecular Cell Science, Chinese Academy of Sciences, 200031 Shanghai, China. [4]Institute of Biotechnology and Health, Beijing Academy of Science and Technology, 100089 Beijing, China. [5]Key Laboratory of Head & Neck Cancer Translational Research of Zhejiang Province, Zhejiang Cancer Hospital, 310022 Hangzhou, Zhejiang Province, China. [6]Zhejiang ShengTing Biotech Co. Ltd, 310018 Hangzhou, Zhejiang Province, China. [7]Shandong Cancer Hospital and Institute, Shandong First Medical University and Shandong Academy of Medical Sciences, 250117 Jinan, Shandong Province, China. [8]Department of Respiratory Medicine, Jinling Hospital, Nanjing University School of Medicine, 210002 Nanjing, Jiangshu Province, China. [9]Department of Respiratory Medicine, Affiliated Hospital of Nanjing University of Chinese Medicine, Jiangsu Province Hospital of Chinese Medicine, 210029 Nanjing, Jiangshu Province, China. [10]Department of Medical Oncology, The First Affiliated Hospital of Wenzhou Medical University, 325000 Wenzhou, Zhejiang Province, China. [11]Department of Medical Oncology, The First Affiliated Hospital, Zhejiang University School of Medicine, 310006 Hangzhou, Zhejiang Province, China. [12]Department of Oncology, Hangzhou Cancer Hospital, 310006 Hangzhou, Zhejiang Province, China. [13]Anhui Province Key Laboratory of Medical Physics and Technology, Institute of Health and Medical Technology, Hefei Institutes of Physical Science, Chinese Academy of Sciences, 230031 Hefei, Anhui Province, China. [14]Hefei Cancer Hospital, Chinese Academy of Sciences, 230031 Hefei, Anhui Province, China. [15]These authors contributed equally: Shirong Zhang, Shutao He, Xin Zhu. ✉e-mail: mashenglin@medmail.com.cn; jtshi@sibcb.ac.cn; gu_hongcang@cmpt.ac.cn

Accurate identification of the primary site is the starting point for cancer diagnosis, and it is critical for guiding the subsequent treatment of metastatic cancer[9,10]. Genetic characterization of cancer of unknown primary sites (CUPs) is also beneficial for some patients. For example, Rassy et al. investigated the clinical response of 234 CUP patients to immune checkpoint inhibitors (ICIs) and concluded that patients with a tumor mutation burden of more than ten mutations per megabase generally have a favorable prognosis[11]. Second, several retrospective studies indicate that 15–20% of CUP patients who receive site-specific chemotherapy have improved overall survival (OS) compared to patients treated with empiric chemotherapy[6,12–14]. The remaining 80–85% do not have a favorable outcome despite identifying the primary tumor site. However, a meta-analysis of 244 CUP trials identified weaknesses in the experimental design of many studies[15]. The authors proposed two comprehensive methods for CUP clinical studies that incorporate the latest detection and treatment options. Therefore, if future clinical trials are conducted accordingly, it is likely that more CUP patients will have an encouraging prognosis[15]. Finally, a definitive diagnosis may spare patients the anxiety or severe psychiatric problems caused by uncertain cancer types[16].

To determine the primary site of metastatic cancer, authoritative organizations have established guidelines consisting of physical checkups, pathological investigations, laboratory tests, and imaging-based studies[4,17]. Among these, immunohistochemistry (IHC) with antibodies against tumor antigens has been the "gold standard" for the past two decades[18]. Yet, the challenges remain: hand-picked antibody panels are primarily subjective, and the IHC analysis can identify primary sites in only 50–65% of patients with metastases and an even lower rate of 20–25% in CUP patients[19,20].

Driven by the hypothesis that metastatic tissues preserve the molecular signatures of primary sites, gene expression-based assays utilizing either RT-qPCR or microarrays have been developed and applied to classify the tissue of origin in metastatic cancer with accuracies ranging from 52.5% to 87%[14,21–25]. Although some tests have been independently validated and applied in the clinic, they generally require samples containing at least 40–80% tumor cells[23–30]. In addition, the ubiquitous presence of RNase further confines the application of RNA-based tests[31], especially when using highly degraded RNA from formalin-fixed, paraffin-embedded (FFPE) samples[32,33].

Compared to the single-stranded RNA, the double-stranded nature and the absence of a reactive 2'-hydroxyl group on the pentose ring make DNA more attractive for genetic testing[34,35]. DNA methylation, the addition of a methyl group to the cytosine almost exclusively in the context of CpG dinucleotides, shows both cell- and tissue-specific patterns in the human genome[36–38]. This feature and characteristic DNA methylation patterns, global hypomethylation and localized hypermethylation, promote the development of DNA methylation-based classifiers to determine the histogenetic origin of cancer[6,39,40]. The classifier (EPICUP) established using DNA methylation microarray data correctly identifies the tissue of origin for 87% of CUP patients[41]. However, the microarray platform is uncommon in diagnostic laboratories and generally requires a large amount of DNA (300 ng) from FFPE tissue, which limits its application[6].

We have previously reported reduced representation bisulfite sequencing (RRBS), a cost-effective method that enriches the CpG-rich portion of the human genome. RRBS covers most promoters, the majority of CpG islands (CGIs), and a reasonable amount of other genomic features[42–44]. In this study, we presented a method specifically designed to generate RRBS libraries using FFPE samples and developed machine learning-based classifiers to predict the primary site of metastatic cancer. The performance of the best classifier was systematically evaluated.

## Results

### Specimen and patient characteristics

To create a comprehensive DNA methylation database and build an appropriate molecular classifier, we excluded samples from 7 primary and 40 metastatic tumor patients due to poor DNA quality or RRBS libraries that failed the quality control (Fig. 1a). Specimens from 27 patients, including 19 with primary tumors and 8 with metastatic tumors, were used for assay development and evaluation. Due to late patient enrollment, RRBS libraries generated from 8 patients (5 with primary tumors and 3 with metastatic tumors) were not included in the downstream analyses. However, all patients with CUP were included in the study regardless of sample and RRBS library quality. Finally, the training set of libraries was constructed using fresh-frozen (FF) tumor tissues from 498 patients with ten common primary cancers, representing 75% and 80% of male and female cancer cases in China, respectively (Table 1; Supplementary Data 1)[45]. The primary tumor tissues were evaluated by experienced pathologists to make sure that each sample had a good representation of tumor cells (60–70%). Genomic DNA was isolated from the FF tissues to ensure DNA integrity, thus guaranteeing the quality of our reference database. Instead of FF samples, the validation samples were FFPE tissues, which preserve the morphological and cellular information of derived tissues[46,47] from 215 patients with metastatic cancer (Table 1, Fig. 1a, and Supplementary Data 1). The criterion for tumor cell content in this cohort was low, 10% or more, to make the test widely applicable in the clinic. Of note, 78 out of 215 samples (36.3%) were biopsied from lymph nodes where tumors of epithelial origin often metastasize (Table 1; Supplementary Data 1)[48].

### Development and validation of the bisulfite sequencing-based assay

Genomic DNA samples isolated from FFPE tissues were highly degraded in almost all cases[49]. RRBS was initially designed to target 40–220 bp fragments to assess genome-wide DNA methylation changes[42,43]. To investigate the feasibility of FFPE samples for methylation profiling, we developed a method called FFPE-RRBS (Fig. 1b). First, the degraded DNA was end-polished by removing the phosphate group from the 5'-terminus so that DNA fragments without MspI digestion would not be included in the sequencing library, a necessary step to ensure that each sequenced fragment contains at least one CpG site[44]. Next, we selected a buffer (CutSmart, NEB, USA) that worked well for dephosphorylation and the downstream reactions. Lastly, all five initial enzymatic reactions were performed sequentially in the same tube without DNA purification. FFPE-RRBS allowed the assay time to be reduced from 6 to 9 days to ~20 h compared to published RRBS methods (Fig. 1b)[43,44].

To evaluate the reliability of FFPE samples for methylation profiling, we generated 19 paired FF- and FFPE-RRBS libraries and compared the data metrics of the two library types (Supplementary Table 1; Fig. 2a–c). The mean read counts were comparable, 37.42 million (M) for the FF-RRBS libraries and 40.61 M for the FFPE-RRBS libraries. In terms of alignment rate, 68.96% (95% CI, 67.75%-70.17%) were mapped to the reference genome for the FF-RRBS libraries compared to 66.74% (95% CI, 64.26%–68.68%) for the FFPE-RRBS libraries (Supplementary Table 1). Bisulfite conversion rates, a parameter to quantify the percentage of unmethylated cytosines correctly converted to uracil, were very high for both (FF, 99.88%; FFPE, 99.63%). However, we observed that the FFPE-RRBS libraries had a narrow size distribution, and the mean insert library sizes were smaller compared to the FF-RRBS libraries (FF: 118 bp; FFPE: 82 bp, $P = 1 \times 10^{-10}$, Fig. 2a and Supplementary Table 1), consistent with a previous report[50]. Interestingly, each of the FF-RRBS libraries ($n = 19$) detected on average about 4.24% of CpGs at ≥5x when randomly sampling 800 K sequencing reads; in contrast, each of the paired FFPE-RRBS ($n = 19$) covered about 3.50% of CpGs at ≥5x ($P = 3.92 \times 10^{-3}$, Supplementary Fig. 1a). Nevertheless, the FFPE-

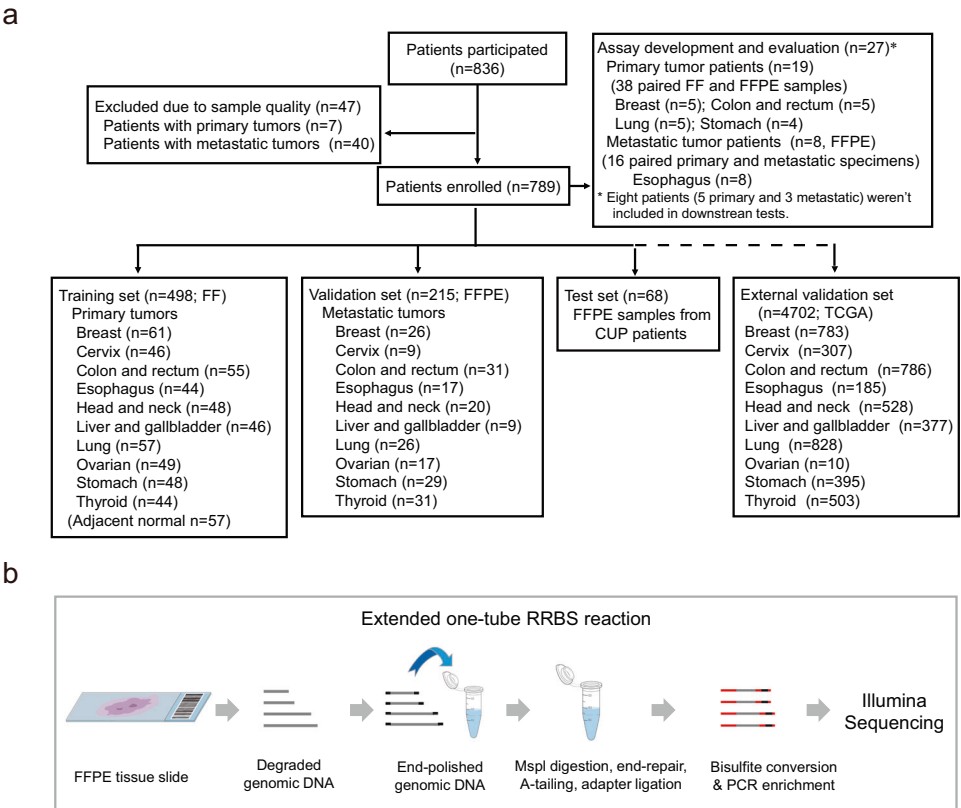

**Fig. 1 | Experimental roadmap of the study. a** Flow chart of participants. **b** Schematic diagram of the FFPE-RRBS protocol. Genomic DNA was isolated from FFPE tissue. The degraded genomic DNA was then end-polished by dephosphorylation. Subsequent enzymatic reactions, including MspI digestion, end-repair, A-tailing, and adapter ligation, were performed in the same tube without DNA cleanup. After bisulfite conversion, the library DNA was enriched by PCR and sequenced on an Illumina NovaSeq 6000 sequencer.

RRBS libraries showed a deeper mean coverage for the CpG sites within CGIs (FF: 42.88x, FFPE: 74.17x; $P = 1.94 \times 10^{-6}$, Fig. 2b), which also translated into a better coverage for CGIs (Fig. 2c). This seems reasonable since RRBS is purposely designed to enrich CpG-rich regions, which may also contain more MspI sites (C|CGG). Therefore DNA degradation has less detrimental effects on CGIs than on CpG-poor regions[42]. In addition, the correlations of the DNA methylation features from all four methylation scoring methods between the paired samples were significantly better than the unpaired samples. The mean methylation of the paired samples demonstrated the highest correlation (Fig. 2d), suggesting that the optimized FFPE-RRBS assay was reliable in capturing the DNA methylation signals.

To explore whether the methylation characteristics of primary cancer reflect those of metastatic cancer, we used paired primary and metastatic cancer tissues to construct 16 FFPE-RRBS libraries. Similar numbers of sequencing reads were obtained, 59.8 M (95% CI, 55.66–63.94M) for primary cancers and 59.41 M (95% CI, 53.05–65.77 M) for metastatic cancers (Supplementary Table 2). The distribution of library sizes, the mean coverage of CpGs across CGIs, and the CGIs with decent coverage all illustrated similar patterns between the library pairs (Fig. 2e–g; Supplementary Table 2). Furthermore, the beta and CHALM values showed better correlations (correlation coefficients ≥0.94) than the other two methylation evaluation methods (Fig. 2h). In addition, we compared the methylation profiles of metastatic tumor tissues (25 liver, 8 lung, and 3 stomach tissues) (Table 1; Supplementary Data 1) with those of the initial tumor sites and the primary tumor occurring at the metastatic sites. The results showed that the correlation between the metastatic tissues and the initial tumor tissues was significantly higher, regardless of how the methylation alterations were calculated (Supplementary Fig. 1b).

We lastly investigated the reproducibility of the FFPE-RRBS assay by selecting 12 samples with a sufficient amount of genomic DNA from 4 cancer types, including lung, breast, liver, and colorectal cancer, and generating triplicate libraries for each sample. As shown in Supplementary Fig. 2, the triplicate libraries exhibited a good correlation with the PCC (Pearson correlation coefficient) values ranging from 0.84 to 0.99, indicating that our FFPE-RRBS protocol is highly reproducible. Together, our data indicated that the strategy of using the FF-RRBS data from primary cancer for model construction and the FFPE-RRBS data of metastatic cancer for validation was logistic and feasible.

## Identifying the primary sites of metastatic cancers

We constructed 28 classifiers by applying the four methylation evaluation methods and seven machine learning approaches to the training set data generated from 498 primary cancer samples. The classifiers were then evaluated by using the validation dataset of 215 FFPE-RRBS libraries that passed the quality control (Supplementary Data 2). The beta value and MHL-based methylation measurements outperformed the other two methods (PDR and CHALM) regardless of the machine learning methods applied, each having 6 out of 7 classifiers with AUC ≥ 0.8 (Fig. 3a and Supplementary Table 3). BELIVE, the beta value-based linear support vector classifier, achieved the best overall performance and was further characterized.

The values of recall, precision, and F1 score for BELIVE exhibited considerable variation among different cancer types, with recall ranging from 0.59 (esophagus) to 0.90 (colon and rectum) and precision varying from 0.52 (head and neck) to 1.00 (colon and rectum) (Fig. 3b; Supplementary Table 4). The F1 scores of five cancer types were greater than 0.80, whereas the two cancer types had relatively lower F1

**Table 1 | Characteristics of primary and metastatic cancer patients**

|  | Metastatic cancer (*n* = 215) | Primary cancer (*n* = 498) |
|---|---|---|
| Cancer types, *n* (%) | | |
| Breast | 26 (12) | 61 (12.2) |
| Cervix | 9 (4.2) | 46 (9.2) |
| Colon and rectum | 31 (14.4) | 55 (11) |
| Esophagus | 17 (7.9) | 44 (8.8) |
| Stomach | 29 (13.5) | 48 (9.6) |
| Head and neck | 20 (9.3) | 48 (9.6) |
| Liver and bile duct | 9 (4.2) | 46 (9.2) |
| Lung | 26 (12) | 57 (11.4) |
| Ovary | 17 (7.9) | 49 (9.8) |
| Thyroid | 31 (14.4) | 44 (8.8) |
| Clinical stage, *n* (%) | | |
| I | 2 (0.9) | 95 (19.1) |
| II | 5 (2.3) | 156 (31.3) |
| III | 49 (22.8) | 159 (31.9) |
| IV | 120 (55.8) | 88 (17.7) |
| Unknown | 39 (18.1) | 0 (0) |
| Biopsy site, *n* (%) | | |
| Lymph node | 118 (54.9) | |
| Liver | 25 (11.6) | |
| Abdominal cavity | 11 (5.1) | |
| Lung | 8 (3.7) | |
| Pelvic cavity | 5 (2.3) | |
| Hydrothorax | 4 (1.9) | |
| Pleura | 4 (1.9) | |
| Chest wall | 3 (1.4) | |
| Neck | 3 (1.4) | |
| Stomach | 3 (1.4) | |
| Others | 30 (14.0) | |
| Unknown | 1 (0.5) | |

values, 0.61 for esophagus cancer and 0.63 for head and neck cancer. The slightly poorer performance was probably due to higher molecular heterogeneity, which is consistent with previous reports[51–53]. Notably, among the correctly predicted samples, most were identified with high confidence (probability >75%), indicating that the prediction was reliable (Fig. 3c). BELIVE achieved an overall prediction accuracy of 81% (Fig. 3e) with an AUC of 0.95 (Fig. 3d). When considering top-*k* accuracy[51], BELIVE achieved a top-3 accuracy of 93% (Fig. 3e) with a median sensitivity of 0.92 across all cancer types (Fig. 3f). Not surprisingly, tumor cell content was positively correlated with prediction accuracy (Fig. 3g). Downsampling analysis of 70 FFPE-RRBS libraries with 24 M reads or more showed that increasing sequencing data could improve the prediction accuracy to some extent (Fig. 3h). To investigate the relationship between sequencing depth and prediction accuracy, we divided FFPE-RRBS libraries (*n* = 69) into low (≤0.5; *n* = 49) and high tumor content groups (>0.5; *n* = 20). Downsampling analysis of both low and high tumor content FFPE-RRBS libraries with 24 M reads or more indicated that increasing sequencing data could also improve the prediction accuracy (Supplementary Fig. 3). Nevertheless, 10 M paired-end reads at 150 bases were sufficient, and 174 (80.9%) FFPE-RRBS libraries met this requirement.

In addition, WGBS has been considered the gold standard for genome-wide methylation profiling and can cover most CGIs. We generated WGBS libraries using 10 metastatic tumor tissues, including breast (*n* = 2), lung (*n* = 1), thyroid (*n* = 1), colon and rectum (*n* = 2), ovary (*n* = 2), esophagus (*n* = 1), and stomach

(*n* = 1) and then performed the correlation analysis by comparing the beta value of CGIs derived from the WGBS and corresponding RRBS libraries. The results showed that all 10 paired libraries had a good correlation with PCC values ranging from 0.87 to 0.96 (Supplementary Fig. 4). Most importantly, BELIVE correctly predicted the primary sites using the WGBS dataset (Supplementary Table 5).

## Validation on an external cohort
To validate BELIVE using the DNA methylation data from The Cancer Genome Atlas (TCGA), we first examined how many CGIs were covered by both RRBS and the Illumina 450 K methylation array. The comparison indicated that 60% (11,353) of the CGIs were in the shared pool (Supplementary Fig. 5). We then used the microarray data from 4702 patients with the 10 most common primary cancers to evaluate the RRBS-based classifier. The test revealed high but variable recall values ranging from 0.81 (lung) to 1.0 (ovarian, liver, and gallbladder) (Fig. 4a; Supplementary Table 6). Precision values were not less than 0.91 for almost all cancer types except for stomach cancer (0.64). In particular, most primary cancers were identified with high confidence (>75% probability) (Fig. 4b). The overall accuracy was 92% with an AUC of 0.99 based on the top-1 prediction, while the accuracy based on the top-3 prediction was as high as 98% with an AUC of 0.99 (Fig. 4c, d; Supplementary Table 6). The median sensitivity across all cancers was 0.99 for the top three predictions (Fig. 4e). In general, the cancers that showed better prediction accuracy using metastatic cancer samples were also highly likely to be correctly identified using primary cancer samples in the TCGA project. For example, the classification of breast, thyroid, colorectal, and rectal cancers all showed an AUC value >0.97 (Supplementary Table 4; Fig. 4c). BELIVE showed a relatively poor performance for esophagus cancer with an AUC of 0.92, probably due to tumor heterogeneity[51–53].

## BELIVE prediction on the tissue of origin for CUP patients
We lastly evaluated the classifier using the FFPE samples from a test cohort of 68 CUP patients whose primary sites were identified by additional pathologic analysis and clinical examination after our testing. BELIVE correctly identified the primary sites in 55 out of 68 patients (~81%) using the top-1 prediction (Table 2; Supplementary Data 3). Moreover, the top-3 prediction achieved an accuracy of approximately 93% (63 out of 68), with the correctly predicted primary sites ranking second in six patients and third in two patients. Of note, one female patient (No. 7) manifested metastases in multiple sites, including the left lower abdomen, retroperitoneal and bilateral inguinal lymph nodes, and bone. The patient was diagnosed with thyroid cancer seven years ago, and the tumor was surgically removed. IHC analysis and H&E staining of two consecutive biopsies from the inguinal mass in November 2019 and June 2020 supported the diagnosis of poorly differentiated adenocarcinoma with signet ring cell carcinoma (Supplementary Fig. 6a). However, neither gastroscopy nor colonoscopy revealed any malignant lesions. Therefore, the tissue of origin remained unknown. Methylation analysis of the second FFPE tissue from the inguinal mass predicted the stomach as the primary site. This conclusion was confirmed by H&E staining and IHC testing of the biopsied tissue from the gastric lesion in September 2020 (Supplementary Fig. 6b), suggesting that BELIVE is a sensitive method for the diagnosis of CUP.

## Discussion
FFPE-RRBS is a streamlined method for DNA methylation profiling using degraded DNA from FFPE tissue. The method applies the CutSmart buffer for five sequential enzymatic reactions and eliminates the DNA cleanup step after each enzymatic reaction. Consequently, the reactions can be performed in a single tube, significantly reducing DNA loss and enabling the generation of RRBS libraries with nanogram DNA

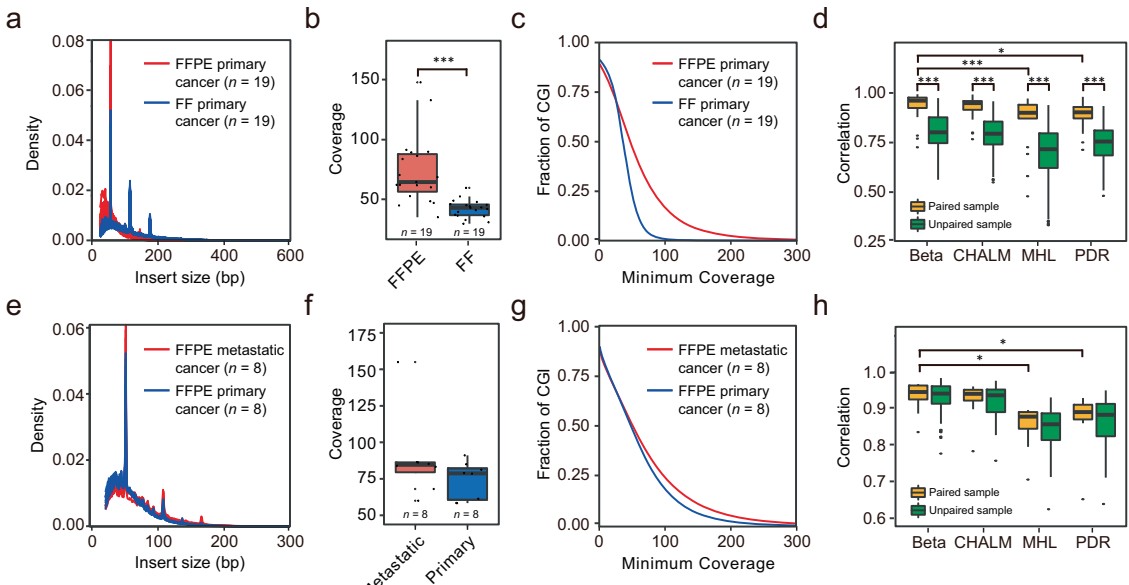

**Fig. 2 | Comparison of RRBS data from different sample types. a–d** RRBS libraries were generated using the paired FF and FFPE tissues from 19 patients with primary cancer. **a** Size distributions for the two library types. FFPE-RRBS (red line), FF-RRBS (blue line). **b** Mean coverage of CpGs within CGIs for the two library types. FFPE-RRBS (red rectangle), FF-RRBS (blue rectangle). **c** Coverage uniformity of CpGs within CGIs. FFPE-RRBS (red line), FF-RRBS (blue line). **d** Comparison of the mean correlation of different DNA methylation metrics for paired (yellow rectangle) and unpaired (green rectangle) sample groups (*n* = 38). DNA methylation of CGIs was assessed using different methods, including beta value, CHALM, MHL, and PDR, as described in the Methods section. **e–h** RRBS libraries were constructed using the FFPE tissues from the primary and matched metastatic tissues of cancer patients (*n* = 8). **e** Size distributions for the FFPE-RRBS libraries derived from the primary (blue line) and matched metastatic (red line) tissues. **f** Mean coverage of

the FFPE-RRBS libraries constructed using the primary (blue rectangle) and matched metastatic (red rectangle) tissues. **g** Coverage uniformity for the RRBS libraries of two tissue groups. Primary tissues (blue line); metastatic tissues (red line). **h** Comparison of the mean correlation of different DNA methylation metrics for paired (yellow rectangle) and unpaired (green rectangle) sample groups (*n* = 16). DNA methylation of CGIs was evaluated using different methods, including beta value, CHALM, MHL, and PDR, as described in the Methods section. In the box plots of **b**, **d**, **f**, and **g**, the center line, box limits and whiskers represent the median, upper and lower quartiles, and 1.5-fold interquartile range, respectively. Asterisks ∗ and ∗∗∗ indicate a significant difference at $P < 0.05$ and $P < 0.001$, respectively, as determined by the two-sided Wilcoxon rank sum test. Source data are available in a supplementary file.

inputs. Data metrics generated from FF- and FFPE-RRBS libraries showed a marked difference in the size distribution due to the nature of FFPE, and the result is consistent with previous reports[46,50]. However, FFPE-RRBS provided a deeper and more uniform coverage of CpGs in CGIs than FF-RRBS, probably because the short DNA fragments from CGIs still have a high chance of preserving two or more MspI sites[54]. Notably, the mean methylation levels of CGIs between paired FF and FFPE-RRBS libraries showed a strong correlation (median correlation of 0.96).

The best classifier, BELIVE, can predict the primary sites with an overall accuracy of 81% with an AUC of 0.95 and a top-3 accuracy of 93% using 215 diverse FFPE tissues, including 36.3% from lymph nodes. Furthermore, BELIVE coincidentally identified the tissue of origin in approximately 81% (55 of 68) and 93% (63 of 68) CUP patients using the top-1 and top-3 prediction methods, respectively. In the real world of cancer diagnosis, the top-*k* accuracy is informative because it helps physicians narrow down the possibilities[51]. Another reason is related to the technical caveat of biopsy, where the tumor cell content is likely to be below the low detection limit; thus, the top-1 prediction corresponds to the biopsied tissue rather than the tissue of cancer origin[55]. Metastatic cancers or CUP may also have multiple primary sites, making the top-*k* predictions more realistic[53]. It is worth noting that BELIVE, trained with FF-RRBS data, is compatible with the methylation array data. Our classifier predicts the primary cancer types of 4702 patients with an overall accuracy of 92%. The prediction accuracy of BLIVE is comparable to the classifier EPICUP, which was trained and validated using only microarray-based methylation data[6]. BLIVE appears to outperform IHC, which relies on manual interpretation and may be limited by the lack of antigens in poorly differentiated cancers[56].

Several features of the test are noteworthy. Our test applies to a wide range of tumor cell contents (10–90%), whereas mRNA- and microRNA-based classifiers mostly require samples with tumor cell contents of at least 40–80%[23–30]. Another robustness of this test is that it only needs a small amount of genomic DNA (10–50 ng). In comparison, the assay using the methylation microarray platform demands a minimum of 300 ng of DNA[6], which is sometimes unrealistic, especially when samples are obtained from fine needle aspirates or core biopsies[57]. The weakness of our study is that the classifier was only established for ten cancer types. However, our method can be easily extended to other cancer types once the samples are available for training and validation.

The treatment and potential outcome of metastatic cancer are largely dependent on the primary site[7,41]. An atypical scenario is that patients with melanoma of unknown primary (MUP), which accounts for approximately 3% of all melanomas, tended to have a better outcome than those with melanoma of known primary (MKP) prior to the era of novel therapy with ICIs[58]. This is also true with the more recent use of immunotherapy, possibly due to increased immunogenicity in patients with MUP. Nevertheless, identifying the tissue of origin is generally required for metastatic cancer patients with uncertain or unknown primary sites. Other scenarios that require identification of the primary site include the development of secondary cancers after cure of the primary cancer and poorly differentiated or undifferentiated cancers. It is estimated that 15% or 16% of cancer patients may need an accurate test to identify the primary sites[6,53]. A widely accepted concept in the medical community is that at least 15–20% of CUP patients have a favorable prognosis with primary site-directed therapy[4]. Typical CUP subgroups with a positive clinical outcome

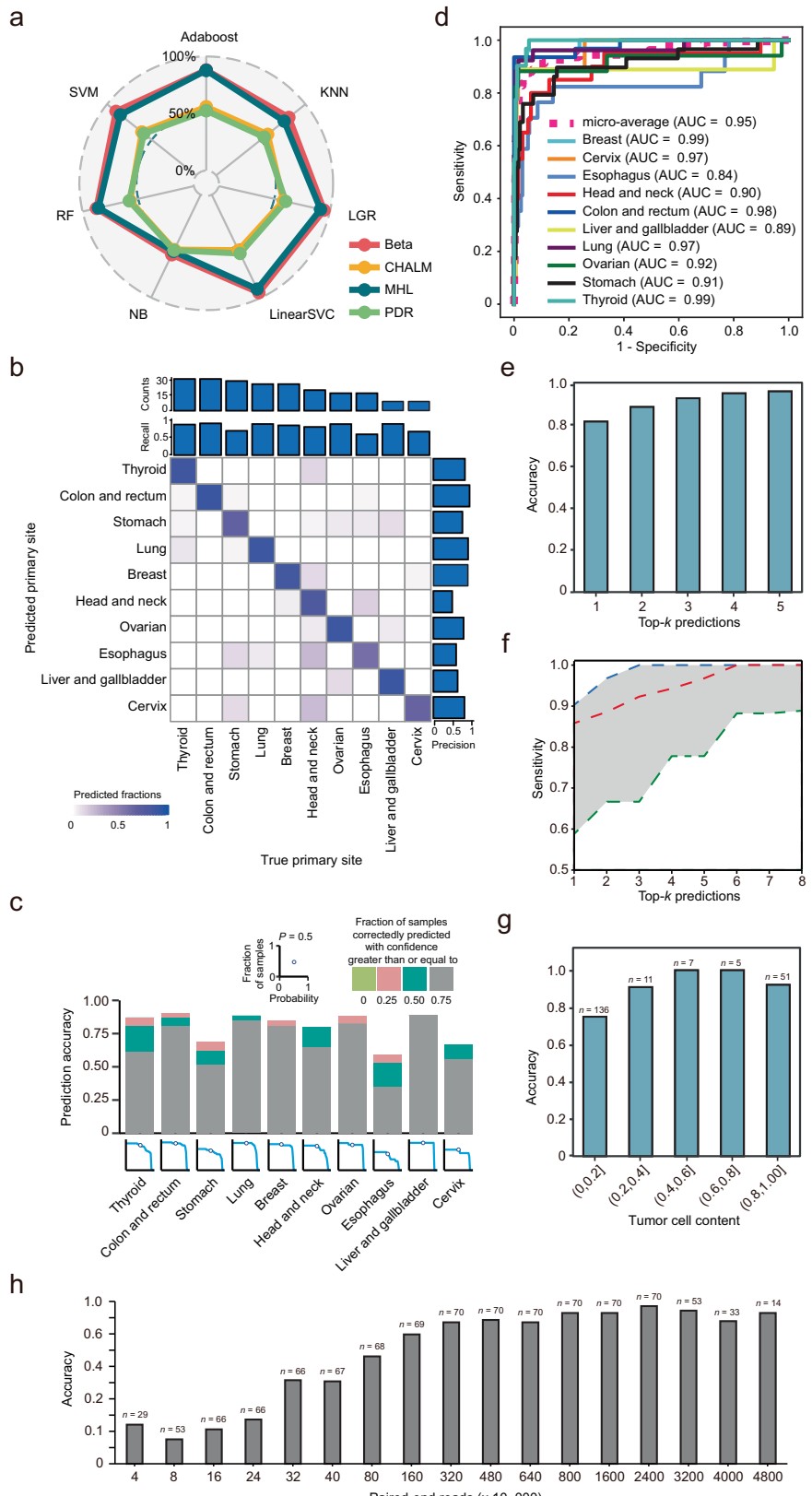

include neuroendocrine carcinoma of unknown primary, squamous cell carcinoma involving cervical lymph nodes, single metastatic deposit of unknown primary, papillary adenocarcinoma of the peritoneal cavity in women, isolated axillary nodal metastases in women, and osteoblastic bone metastases and prostate-specific antigen expression in men. Excitingly, additional subgroups of CUP with better

prognoses, such as colorectal, renal, and pulmonary, have recently emerged and may benefit from tailored treatment plans[59]. With the increasing number of new treatment options, such as targeted therapies and immunotherapies, the test that can accurately identify the primary site of cancer will benefit thousands of cancer patients[10,21,60]. In particular, there is emerging evidence that certain genetic features,

**Fig. 3 | Comparison of 28 classifiers and the performance of BELIVE in predicting the primary sites of metastatic cancer (*n* = 215). a** The radar plot illustrates the area under the curve (AUC) values of 28 classifiers. **b** BELIVE performance in detecting the primary sites of ten common metastatic cancers. Sample size and recall are plotted at the top of the confusion matrix, while precision is plotted on the right. The rows in the matrix show the primary cancer sites predicted by BELIVE and the columns show the authentic primary cancer sites. The colored squares along the diagonal represent the percentage of primary cancers correctly identified by BELIVE. **c** The bar chart (top) shows the proportion of samples whose primary sites were correctly identified with different confidence levels; the area charts (bottom) show the proportion of samples (*y*-axis) whose primary sites of metastatic cancers were correctly classified with greater than or equal to a confidence level

(*x*-axis). **d** ROC curves for the classification of primary metastatic sites. **e** Top-*k* accuracies for predicting primary sites of metastatic cancers. **f** Sensitivities of BELIVE based on top-*k* predictions. The red line shows the median sensitivity of BELIVE for predicting primary sites across the ten cancers, while the blue and green lines correspond to the sensitivities for the best and worst performing cancers. **g** Prediction accuracies of BELIVE for different bins of tumor cell content. **h** Prediction accuracies were calculated over different inputs of sequencing data. The top 70 FFPE-RRBS libraries with more than 24 million paired-end reads were subjected to a downsampling analysis. After randomly dropping a fraction of the sequencing reads, the remaining data was used to test BELIVE's prediction accuracy. Of the 70 libraries, 53 with 32 million or more sequencing reads were also evaluated. Source data are available in a supplementary file.

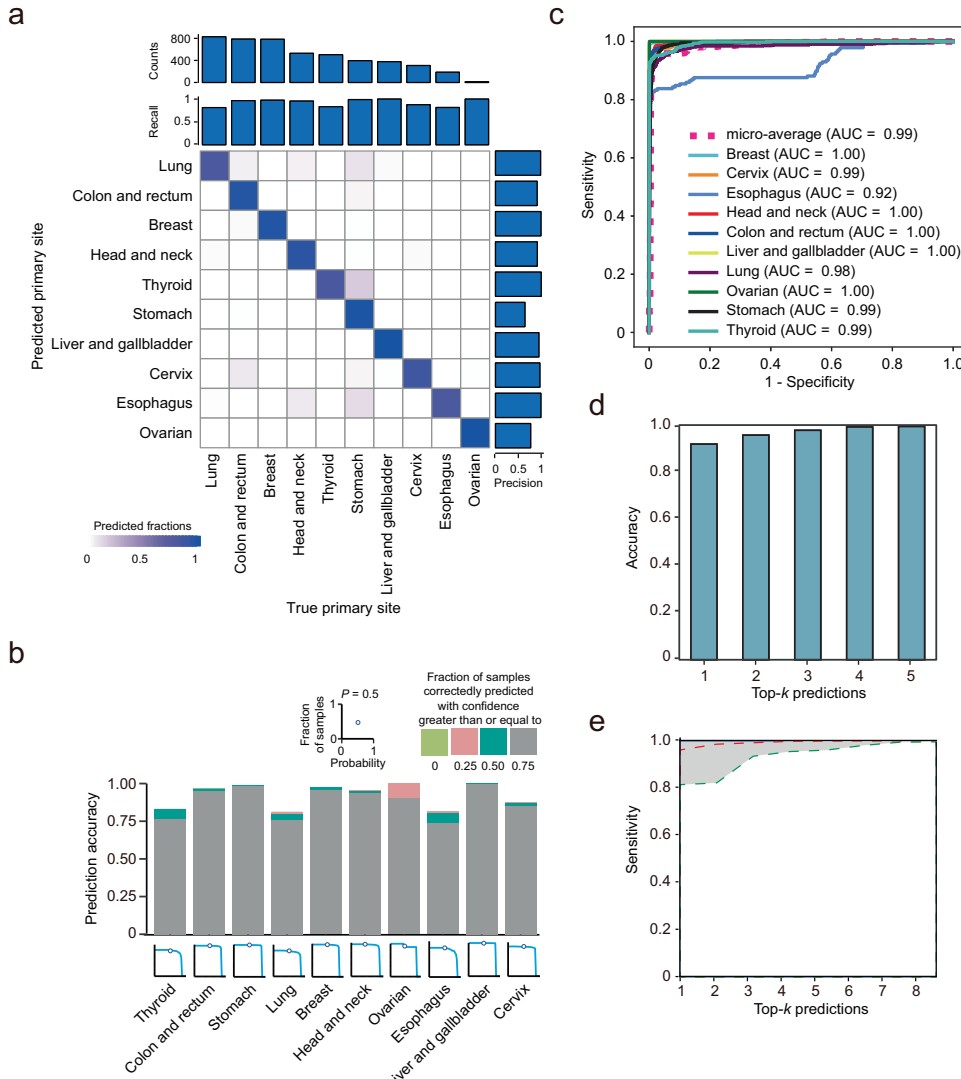

**Fig. 4 | Performance of the BELIVE algorithm on the TCGA DNA methylation microarray dataset (*n* = 4702). a** BELIVE performance in tissue of origin detection for patients diagnosed with primary cancers in the TCGA project. Sample size and recall are plotted at the top of the confusion matrix, while precision is plotted on the right. The columns in the matrix show the primary cancer sites predicted by BELIVE, and the rows show the actual sites. Colored squares along the diagonal represent the percentage of primary sites correctly identified by BELIVE. **b** The bar graph (top) shows the proportion of samples whose primary sites were correctly identified at different confidence levels; the area plots (bottom) show the

proportion (*y*-axis) of samples whose primary sites were correctly classified with greater than or equal to a confidence level (*x*-axis). **c** ROC curves for prediction of primary sites in patients with primary cancer. **d** Top-*k* (*k* = 1, 2, 3, 4, 5) accuracies for predicting primary sites in patients with primary cancer. **e** Sensitivities of the BELIVE algorithm based on the top-*k*-ranked predictions. The red line shows the median sensitivity of BELIVE in predicting the primary sites of ten cancers, while the blue and green lines correspond to the sensitivities for the best and worst performing cancers. Source data are available in a supplementary file.

**Table 2 | BELIVE prediction accuracy in 68 CUP patients**

| Diagnosed cancer type | Patients (n) | Top-1 Prediction (n) | Top-1 Accuracy | Top-3 Prediction (n) | Top-3 Accuracy |
|---|---|---|---|---|---|
| Lung | 19 | 18 | 95% | 18 | 95% |
| Head and neck | 17 | 12 | 71% | 16 | 94% |
| Stomach | 8 | 8 | 100% | 8 | 100% |
| Colorectum | 7 | 4 | 57% | 6 | 86% |
| Ovary | 5 | 4 | 80% | 4 | 80% |
| Liver and bile duct | 6 | 6 | 100% | 6 | 100% |
| Cervix | 2 | 1 | 50% | 2 | 100% |
| Thyroid | 3 | 1 | 33% | 2 | 67% |
| Esophagus | 1 | 1 | 100% | 1 | 100% |
| Total | 68 | 55 | 81% | 63 | 93% |

such as chromosomal instability (CIN) and specific gene mutations, can be used as biomarkers to guide the use of ICI in CUP patients. However, clinical trials are still needed to investigate the relationship between CIN, gene mutations, and immunotherapies[61].

In conclusion, FFPE-RRBS is a reproducible and streamlined method for DNA methylation profiling using degraded DNA fragments isolated from FFPE samples. Based on RRBS data, BELIVE can effectively identify the tissue of origin for metastatic cancer and CUP patients. Incorporating additional cancer types into the classifier will further expand its clinical application, which is currently undergoing extensive evaluation.

## Methods

This study was approved by the institutional review board (IRB) of the participating hospitals, including Hangzhou First People's Hospital, Zhejiang Cancer Hospital, Shandong Cancer Hospital and Institute, and Hefei Cancer Hospital of the Chinese Academy of Sciences. Written informed consent was obtained from the patients. Our investigation was conducted in accordance with all relevant ethical regulations.

### Patients and samples

Participants enrolled in this study included: (1) two retrospective cohorts of patients diagnosed with either primary cancer (n = 510) or metastatic cancer (n = 258) between April 2015 and July 2021; (2) a prospective cohort of patients (n = 68) initially classified as having CUP between July 2021 and April 2023 (Fig. 1a). Criteria for diagnosis of CUP were based on NCCN guidelines[17]. Patient eligibility criteria are described in the Supplementary Materials and Methods.

In total, we collected 510 FF tumor samples, 57 adjacent normal tissues, and 258 FFPE tissues from metastatic cancer patients for assay development and testing. The study also included 68 FFPE samples from CUP patients with a definitive diagnosis at the end of the study. Medical histories were obtained for all patients. Board-certified pathologists re-evaluated tumor histology. DNA methylation microarray (Illumina 450 K) data of the ten most common cancers from the TCGA project were downloaded and used for cross-validation (n = 4702).

### Immunohistochemistry analysis

IHC staining and analysis were performed using a combination of antibodies as previously described[62]. The IHC panel in this study included antibodies against Arg-1, Bcl-2, Bcl-6, CA IX, CA125, CAM5.2, CD10, CD117, CD20, CD21, CD3, CD34, CD45, CD56, CDX-2, CEA, c-erbB-2, CgA, CK, CK(HMW), CK18, CK19, CK20, CK5/6, CK7, CK8/18, CR, Desmin, E-cadherin, EGFR, EMA, ER, Galectin-3, GATA-3, Glut-1, GPC-3, GS, Hepatocyte, HSP70, IMP3, INI-1, MUC-1, MUC5AC, MUC-6, MUM-1, Napsin-A, P16, P40, P504s, P53, P63, Pax5, Pax-8, PR,

SATB2, SMARCA4, Syn, TFE3, TG, TTF-1, Villin, Vimentin and WT-1 (Supplementary Data 4). All IHC images were independently reviewed by pathologists.

### RRBS library preparation and sequencing

The assay was conducted using genomic DNA purified from either 10–20 mg of FF tissue or 5–8 of 5–10 μm FFPE tissue sections. Board-certified pathologists reviewed an H&E stained slide to ensure that tumor cells accounted for 10% or more of the cell population and that the necrosis area was less than 50%. Genomic DNA was isolated from FF tissues and FFPE sections using the TIANamp Genomic DNA Kit (TIANGEN, Beijing, China) and TIAamp FFPE DNA Kit (TIAGEN, Beijing, China), respectively, per the manufacturer's recommendations.

Genomic DNA (10–50 ng) from FFPE samples was treated with 0.5 units of shrimp alkaline phosphatase (rSAP, New England Biolabs, USA) in l5 μl of 1x CutSmart buffer (New England Biolabs, USA) at 37 °C for 50 min. The rSAP was then inactivated at 75 °C for 20 min. Dephosphorylated DNA was treated with 5 units of MspI (New England Biolabs, USA) at 37 °C for 90 min in a final reaction of 16 μl, followed by heat-inactivation of the restriction enzyme at 70 °C for 10 min. The digested DNA was end-repaired and A-tailed in an 18 μl reaction containing 2.5 units of Klenow enzyme (3′–5′ exo-New England Biolabs, USA), 0.2 mM dATP, 40 nM dCTP and 40 nM dGTP (New England Biolabs, USA). The reaction was incubated at 30 °C for 25 min, 37 °C for 25 min, and heat-inactivated at 70 °C for 10 min. Barcoded methylated adaptors (0.1 μM) were then ligated to the dA-tailed DNA fragments in a 21 μl reaction containing 0.5 mM ATP, 80 units of T4 ligase (New England Biolabs, USA), and 1× CutSmart buffer at 16 °C for at least 3 h. The T4 DNA ligase was then heat-inactivated at 70 °C for 15 min. Adapter-equipped DNA fragments were purified using 1.6× SPRI beads (Agencourt AMPure XP, Beckman Coulter) and then eluted with 40 μl H2O. The eluted DNA was subjected to sodium bisulfite conversion according to the manufacturer's recommendations (Qiagen, Germany). Bisulfite converted DNA was PCR amplified using primers consisting of Illumina i7 and i5 indexes and thermocycler conditions were 98 °C for 1 min, then 6 cycles of (98 °C for 20 s, 58 °C for 30 s, 72 °C for 1 min) followed by 12 cycles of (98 °C for 20 s, 65 °C for 30 s, 72 °C for 1 min), then 72 °C for 2 min followed by 4 °C hold. PCR products were purified using 1.5x SPRI beads and quantified using an Agilent 2100 Bioanalyzer. FFPE-RRBS libraries were subjected to 150 cycles of paired-end sequencing runs on a NovaSeq 6000 platform (Illumina, USA) with a 30% PhiX spike-in. For FF tissues, RRBS libraries were constructed using 10–20 ng DNA without rSAP treatment[63], but the libraries were sequenced as described above.

### RRBS data processing

Approximately 8 gigabases of RRBS data were obtained for each library. Adapter and barcode sequences were removed using Trim Galore (version 0.6.2)[64]. The trimmed reads were mapped to the human genome version hg19 using BSMAP[65], with the options "-q 20 -f 5 -r 0 -v 0.05 -s 16 -S 1". The resulting BAM files were then converted to mHap files using the mHapTools (version 1.0)[66]. CpG methylation metrics were extracted using the MethylDackel tool developed by Devon Ryan [https://github.com/dpryan79/MethylDackel]. We filtered out samples with low bisulfite conversion rates (<99%), low mapping ratios (<50%), and insufficient numbers of CpGs (<0.8 million) at 10x coverage. Consequently, RRBS libraries from 498 primary tumors, 215 metastatic tumors, and 68 CUP patients were subjected to further analysis.

### Feature selection and classifier construction

We first computed the methylation levels of CGIs using four methods, including mean methylation (beta value)[67], proportion of discordant reads (PDR)[68], cell heterogeneity-adjusted clonal methylation (CHALM)[69], and methylated haplotype load (MHL)[70]. Only CGIs with coverage >100x were further analyzed. Next, we applied the two-sided

Wilcoxon rank sum test to select CGIs that could be employed as biomarkers from the training dataset of 498 FF primary tumors and 57 adjacent normal tissues. Noteworthily, the CGIs should meet the following conditions: (1) The methylation level of a selected CGI in one cancer type should show a significant difference compared to other cancer types (FDR ≤ 0.01). For this purpose, we calculated the methylation level of each CGI in one cancer type, compared it with that in the remaining nine cancer types, and selected only the cancer type-specific CGI, regardless of whether it is hypomethylated or hypermethylated. (2) The selected CGI should also have a significantly different methylation level (FDR ≤ 0.01) compared to that in the control group of 57 tumor-adjacent normal tissues. The code for the selection of CGIs is available at https://github.com/heshutao0420/cup.

Thus, 28 classifiers were developed by using four sets of selected biomarkers and categorizing the biomarkers of different cancer types using the Scikit-learn package, which included seven machine learning approaches, including AdaBoost, k-nearest neighbor (KNN), logistic regression (LGR), linear support vector classifier (LinearSVC), Naïve Bayesian (NB), random forest (RF), and support vector machine (SVM). Specifically, we used a Bayesian optimization approach to select the hyperparameters and then divided the 498 FF-RRBS libraries into training and validation datasets according to the ratio of 4:1. To balance the number of RRBS libraries in the training and validation sets, the SMOTE function from the imblearn Python library was employed. For the training set, we used the hyperopt Python library to select the following hyperparameters: (1) Adaboost: the maximum number of estimators at which the boosting stops, the learning rate; (2) KNN: the number of neighbors, the algorithm used to compute the nearest neighbors, and the weight function used in the prediction; (3) LGR: the penalty, the inverse of the regularization strength, the algorithm used in the optimization problem; (4) LinearSVC: norm used in penalization, loss function, tolerance for stopping criteria, regularization parameters, multi-class strategy, the maximum number of iterations to run; (5) NB: additive (Laplace/Lidstone) smoothing parameter; (6) RF: maximum depth of the tree, number of features to consider when searching for the best split, number of trees in the forest, feature selection criteria; (7) SVM: regularization parameter, kernel type to be used in the algorithm, kernel coefficient. After optimizing the hyperparameters, the prediction probability of each model was calibrated using the CalibratedClassifierCV function from the Scikit-learn package on the corresponding validation set. The codes are available at https://github.com/heshutao0420/cup.

These classifiers were evaluated by using the methylation features extracted from the validation dataset of 215 FFPE tissues, and only the best one was further evaluated. Quantitative metrics, including precision, recall, F1 score and accuracy, were used to evaluate the performance of the classifiers. TP (true positive), TN (true negative), FP (false positive), and FN (false negative) were calculated based on the confusion matrix. The formulas used to calculate the quantitative metrics were presented as follows:

$$\text{Recall} = TP/(TP + FN). \tag{1}$$

$$\text{Precision} = TP/(TP + FP). \tag{2}$$

$$F1 = 2(\text{recall} * \text{precision})/(\text{recall} + \text{precision}). \tag{3}$$

$$\text{Accuracy} = (TP + TN)/(TP + FP + TN + FN). \tag{4}$$

## Reporting summary
Further information on research design is available in the Nature Portfolio Reporting Summary linked to this article.

## Data availability
The RRBS and WGBS datasets generated in this study were deposited in the NCBI Gene Expression Omnibus (GEO) under accession code GSE231984, which included the following SubSeries: training set (accession code: GSE230193), validation set (accession code: GSE231969), CUP test set (accession code: GSE233087), WGBS set (accession code: GSE233088). The processed data in bedGraph and mHap format are freely available at GEO, which is sufficient to reproduce the results in this study. We have deposited the raw data in the Genome Sequence Archive (GSA) under the accession number HRA005166 under controlled access. Access can be requested through Hongcang Gu (gu_hongcang@cmpt.ac.cn) and will be made available for non-commercial use for a minimum of 5 years. The DNA methylation microarray (Illumina 450 K) data of the ten most common cancers used in this study are available in the TCGA database. The source data generated in this study were recorded in the source data files. Source data are provided with this paper.

## Code availability
The script was implemented using Python 3.6.11 and R 4.0.0. Other tools and packages used for data analysis include: numpy 1.19.1, pandas 1.1.5, scipy 1.5.2, sklearn 0.23.3, imblearn 0.0, hyperopt 0.2.5, argparse 1.1, matplotlib 3.5.2, glob2, ggplot2 3.3.5, and GenomicRanges 1.34.0. The codes used in this study are available from *Zenodo* [https://zenodo.org/record /8022705][71]. The codes are maintained and updated at https://github.com/heshutao0420/cup.

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

## Acknowledgements

The research was supported by the following agencies: Key Program Cosponsored by Zhejiang Province and National Health Commission of China (Grant ID: WKJ-ZJ-2331, to S.Z.), National Natural Science Foundation of China (Grant ID: 82072950, X.Z.), National Natural Science Foundation of China (Grant ID: 32270691 to J.S.), Major Project of Hangzhou Science and Technology Bureau (Grant ID: 20180417A01, to S.M.). J.S. is a recipient of the Hundred Talents Program Award of the Chinese Academy of Sciences. H.G. is supported by the CASHIPS seed grant. This work was supported by Westlake Laboratory (Westlake Laboratory of Life Sciences and Biomedicine).

## Author contributions

S.Z., S.M., J.S., H.G., and Y.W. designed the study. S.H., X.Z., Q.X., C.-Q.X. contributed to data analysis. S.Z., S.M., X.Z., C.-W.X., W.W., X.S., L.X., J.Y., W.L., X.-C.Z., Q.W., and Y.W. were responsible for patient recruitment. S.Z., S.H., H.G., J.S., S.M. drafted the manuscript. All authors participated in drafting or revising the manuscript.

## Competing interests

Q.X., Y.W., and C.X. are full-time employees of Zhejiang ShengTing Biotech. H.G. is a co-founder of Zhejiang ShengTing Biotech. All other authors reported no competing interests.
