## [Peer Review File · Nature Communications]

DNA methylation profiling to determine the primary sites of metastatic cancers using formalin-fixed paraffin-embedded tissuesRESPONSE TO REVIEWERS' COMMENTS

Reviewer #1 (Remarks to the Author): clinical expertise in cancer of unknown primary

In this article, the authors present a novel method specifically for generating RRBS libraries using FFPE samples and develop machine learning-based classifiers for predicting the primary site of metastatic cancers. The manuscript is straightforward, well written, and concise and has clear results. Definitely deserves to be published and is a valuable contribution to the "Nature Communications" journal. Some comments need to be addressed before publication.

We are honored that such a distinguished expert as this reviewer has read our manuscript and made excellent suggestions from a clinician's point of view. The kind words regarding our manuscript are really encouraging for all the authors. Over the past few years, our team has been striving to develop new NGS-based sequencing technologies, investigate different algorithms, and enroll patients in this project. We are happy to present our findings to the biomedical community and sincerely hope our results will benefit CUP patients.

[1] "INTRODUCTION," Lines 69-70:

"Accurate identification of the primary site is the starting point for cancer diagnosis, and it is critical to guide the subsequent treatments of metastatic cancer [9, 10]."

In that regard, it is worthy to be reported that 28% of patients with CUP present one or more predictive biomarkers to immune checkpoint inhibitors (ICI), such as programmed death-ligand 1 (PD-L1) expression on $\geq 5\%$ cancer cells in 22.5% ($\geq 1\%$ in 34%) and lymphocytes in 58.7%, microsatellite instability (MSI)-high in 1.8% and tumour mutational burden (TMB) ≥ 17 mutations per megabase in 11.8%. However, these biomarkers are not yet validated in patients with CUP. Generally, CUP patients with TMB > 10 mutations per megabase have a trend for better outcomes when treated with ICI.

Recommended reference: Rassy E, et al. Genomic correlates of response and resistance to immune checkpoint inhibitors in carcinomas of unknown primary. *Eur J Clin Invest.* 2021;51(9): e13583.

We thank the reviewer for mentioning this excellent reference. The conclusion fits well with the content of this manuscript, so we are happy to cite the reference in our revised manuscript. Here is what we stated:

Accurate identification of the primary site is the starting point for cancer diagnosis, and it is critical to guide the subsequent treatment of metastatic cancer [9, 10]. *Genetic characterization of CUP is also beneficial for some patients. For example, Rassy et al. investigated the clinical response of 234 CUP patients to immune checkpoint inhibitors (ICIs) and concluded that patients with a tumor mutation burden (TMB) of more than ten mutations per megabase generally have a favorable prognosis[11]. (Lines 70-74).*

[2] "INTRODUCTION", Lines 70-73:

"Multiple retrospective studies indicate that CUP patients who receive site-specific chemotherapy have improved overall survival (OS) compared with patients treated with empirical chemotherapy [6, 11-13]."

At that point, the authors are highly encouraged to report that there are significant deficiencies in the currently available studies comparing site-specific therapy and empiric chemotherapy. These deficiencies include patient accrual problems (oversampling treatment-resistant tumor types and long recruitment), study design limitations (observational and problematic trials), heterogeneity among the CUP classifiers (epigenetic vs. Transcriptomic profiling), and incomparable therapies. The assessment of recently published CUP literature allows to recommend two comprehensive clinical trial designs, a visionary and a pragmatic approach. Both are amenable to implementing the latest diagnostics and therapeutic advances to improve the quality of CUP research and the prognosis of many patients.

Recommended reference: Rassy E, et al. Systematic review of the CUP trials characteristics and perspectives for next-generation studies. *Cancer Treat Rev.* 2022; 107:102407.

We agree with the reviewer. There are limitations to many studies that assess whether site-specific treatment is better than empiric treatment. We have changed our tone slightly and added the critical point as follows:

Retrospective studies indicate that 15-20% of CUP patients who receive site-specific chemotherapy have improved overall survival (OS) compared with patients treated with empirical chemotherapy [6, 11-13]. *The remaining 80-85% do not have a favorable outcome despite identifying the primary tumor site. However, a meta-analysis of 244 CUP studies identified weaknesses in the experimental design of many trials [15]. The authors proposed two comprehensive methods for CUP clinical studies that incorporate the latest detection and treatment options. Therefore, when future clinical trials are conducted accordingly, it is likely that more CUP patients will have an encouraging prognosis [15]. (Lines 77-82).*

[3] "DISCUSSION", Lines 371-372:

"The treatment and potential outcome of metastatic cancer largely depend on the primary site [7, 39]."

The authors are strongly recommended to mention the case of melanoma of unknown primary (MUP), which represent approximately 3% of all melanomas. Recently has been published that patients with MUP site seem to present better outcomes compared to those with stage-matched melanoma of known primary (MKP), probably due to higher immunogenicity as reflected in the immunologically mediated primary site regression. As such, MUP patients on immunotherapy probably display better outcomes when compared to the MKP site subset.

Recommended reference: Boussios S, et al. Melanoma of unknown primary: New perspectives for an old story. *Crit Rev Oncol Hematol.* 2021; 158:103208.

We thank the reviewer for pointing out this excellent review. The updated description can

be found on lines 440-444 and is shown below:

The treatment and potential outcome of metastatic cancer largely depend on the primary site [7, 39]. *An atypical scenario is that patients with melanoma of unknown primary (MUP), which accounts for about 3% of all melanomas, tended to have a better outcome than those with melanoma of known primary (MKP) prior to the era of novel therapy with ICIs [67]. This is also true with the more recent use of immunotherapy, possibly due to increased immunogenicity in patients with MUP. (Lines 440-444).*

[4] "DISCUSSION", Lines 376-378:

"In the medical community, a widely accepted concept is that at least 15-20% of CUP patients have a favorable prognosis under the primary site-guided therapy [4]."

Please, do report that the favorable risk cancer subgroup includes patients with neuroendocrine carcinomas of unknown primary, peritoneal adenocarcinomatosis of a serous papillary subtype, isolated axillary nodal metastases in females, squamous cell carcinoma involving non-supraclavicular cervical lymph nodes, single metastatic deposit from unknown primary and men with blastic bone metastases and PSA expression. Very recently, new favorable subsets of CUP seem to emerge including colorectal, lung and renal CUP which underlies specific treatments.

Recommended reference: Rassy E, et al. New rising entities in cancer of unknown primary: Is there a real therapeutic benefit? *Crit Rev Oncol Hematol.* 2020; 147:102882.

We strongly agree with the reviewer to include the subsets of CUP with a favorable prognosis. Readers should have a better understanding of the current clinical status of CUP. The new statement will be presented in the revised manuscript as follows:

In the medical community, a widely accepted concept is that at least 15-20% of CUP patients have a favorable prognosis under primary site-guided therapy [4]. *Typical CUP subgroups with a positive clinical outcome include neuroendocrine carcinoma of unknown primary, squamous cell carcinoma involving cervical lymph nodes, single metastatic deposit of unknown primary, papillary adenocarcinoma of the peritoneal cavity in women, isolated axillary nodal metastases in women, and osteoblastic bone metastases and PSA expression in men. Excitingly, additional subgroups of CUP with a better prognosis have recently emerged, such as colorectal, renal, and pulmonary, which may benefit from tailored treatment plans [68]. (Lines 450-456).*

[5] "DISCUSSION", Lines 378-380:

"With increasing new treatment options, such as targeted therapies and immunotherapies, the test that can accurately identify cancer primary site will benefit thousands of cancer patients [10, 19, 64]."

Furthermore – from the therapeutic point of view – chromosomal instability (CIN) is not a frequent phenomenon in CUP, which may favour immune checkpoint inhibitors (ICI) among patients with CUP. Conversely, these patients present individual gene alterations implicated in immune-evasion and resistance to ICI. Further clinical investigations are needed to provide more information regarding the interplay between CIN, point mutations

and the immune system, allowing a better understanding of ICI use in patients with CUP and potentially improving their efficacy.

Recommended reference: Chebly A, et al. Chromosomal instability in cancers of unknown primary. *Eur J Cancer*. 2022; 172:323-325.

We are grateful to the reviewer for providing the reference and the critical point, which perfectly aligns with our position. The relevant information has been incorporated into the revised manuscript as follows:

With increasing new treatment options, such as targeted therapies and immunotherapies, the test that can accurately identify cancer primary site will benefit thousands of cancer patients [10, 19, 64]. *In particular, there is emerging evidence that certain genetic features, such as chromosomal instability (CIN) and specific gene mutations, can be used as biomarkers to guide the use of ICI in CUP patients. However, clinical trials are still needed to investigate the relationship between CIN, gene mutations and immunotherapies [70].* (Lines 458-462).

Reviewer #2 (Remarks to the Author): technical expertise in methylation sequencing
Identification of primary tumour site after detection of metastasis can be challenging. These cases are defined as cancers of unknown primary (CUP). CUP complicates treatments, which are often tailored to the origin of tumours. In this manuscript Zhang et al aimed to develop DNA methylation-based classifier, which would identify primary site of cancer. First, authors adapted reduced representation bisulfite sequencing method (RRBS) to be applied to the formalin fixed paraffin-embedded tissues (FFPE). DNA methylation reference data was collected from the cohort of 498 fresh frozen primary tumour samples and validation cohort was extracted from 215 FFPE samples. Beta value-based linear support vector classifier (BELIVE) demonstrated the best performance. Finally, authors employ the developed classifier to detect primary tumour sites on 33 CUP patients with 88% accuracy.

While the manuscript presents potentially important advancements in identifying cancer origin in CUP patients, the manuscript lacks details, which would enable the study to be reproduced and broadly used by a biomedical community. Specific criticisms are as follows:

We would like to take this opportunity to express our sincere gratitude to the reviewer for evaluating our manuscript. The insightful comments and suggestions are invaluable and have helped us improve the quality of our work. The critical evaluation and constructive feedback have enabled us to identify and address several areas of weakness. In the last three months, we have conducted several experiments, and fortunately, all the results seem to support the method we have developed and the model we have built.

We have responded to each of these as set out below:

1. In the abstract, authors claim to present a novel DNA methylation detection method, however, there is not enough details presented, which would enable evaluation of the novelty of the method. In materials and methods, there is a superficial description of RRBS

with reference to paper by Boyle P et al (2012), which describes much longer version of the protocol. In the main text, authors mention the dephosphorylation before MspI digestion and the choice of the buffer. Authors must present all the details (in materials and methods part), including enzyme manufacturers, concentrations, incubation times, etc., such that the community would be able to benefit from the methodological discoveries.

We thank the reviewer for pointing out the weakness of the library construction procedure. As the reviewer mentioned, the community may also be interested in how we perform the assay step by step. Therefore, we provide a detailed description in our revised manuscript as follows:

Genomic DNA (10-50 ng) from FFPE samples was treated with 0.5 units of shrimp alkaline phosphatase (rSAP, New England Biolabs, USA) in 15 µl of 1x CutSmart buffer (New England Biolabs, USA) at 37°C for 50 min. The rSAP was then inactivated at 75°C for 20 minutes. Dephosphorylated DNA was treated with 5 units of MspI (New England Biolabs, USA) at 37°C for 90 minutes in a final reaction of 16 µl, followed by heat-inactivation of the restriction enzyme at 70°C for 10 minutes. The digested DNA was end-repaired and A-tailed in an 18 µl reaction containing 2.5 units of Klenow enzyme (3'-5' exo-New England Biolabs, USA), 0.2 mM dATP, 40 nM dCTP and 40 nM dGTP (New England Biolabs, USA). The reaction was incubated at 30°C for 25 min, 37°C for 25 min, and heat-inactivated at 70°C for 10 min. Custom barcoded methylated adaptors (0.1 µM) were subsequently ligated to the dA-tailed DNA fragments in a 21 µl reaction containing 0.5 mM ATP, 80 unites of T4 ligase (New England Biolabs, USA), and 1x CutSmart buffer at 16°C for a minimum of 3 hours. Next, the T4 DNA ligase was heat-inactivated at 70°C for 15 minutes. Adapter-equipped DNA fragments were purified using 1.6x SPRI beads (Agencourt AMPure XP, Beckman Coulter) and then eluted with 40 µl H₂O. The eluted DNA was subjected to sodium bisulfite conversion according to the manufacturer's recommendations (Qiagen, Germany). Bisulfite-converted DNA was PCR amplified, and the PCR products were purified as previously reported [46]. (Lines 159-175).

2. The RRBS data must be made available in publicly accessible data repository such a short read archive (and accession number indicated).

We thank the reviewer for this reasonable requirement. All team members are also happy to share our data with the community. We have added a paragraph to our revised manuscript and provided the accession number (GSE231984, secure token for reviewers: ytefcqweztixlon) accordingly. (Lines 487-496).

Data availability

RRBS and WGBS datasets have been deposited in the NCBI Gene Expression Omnibus under accession number GSE231984 (secure token for reviewers: ytefcqweztixlon). (Lines 487-490).

Code availability

The script was implemented using Python 3.6.11 and R 4.0.0. Other tools and packages used for data analysis include: numpy 1.19.1, pandas 1.1.5, scipy 1.5.2, sklearn 0.23.3, imblearn 0.0, hyperopt 0.2.5, argparse 1.1, matplotlib 3.5.2, glob2, R 4.0.0, ggplot2 3.3.5, and GenomicRanges 1.34.0. Codes are available from <https://github.com/heshutao0420/cup>. (Lines 492-496).

3. In Fig 3g the first bar should be labelled 0.1 to 0.2 based on what is indicated in text (>10% tumour content). I find it very surprising that the accuracy does not correlate positively with tumour content. Is tumor content reflected in sequencing data as it is in the histological samples? What is relationship between sequencing depth (read numbers) and accuracy in low and high tumour contents groups?

We thank the reviewer for the excellent question and thought the reviewer's point was more logical. So we checked our lab notes and found that the sizes of metastatic tumor tissue for H&E staining were mostly small, especially considering that more than a third of the samples were derived from lymph nodes. In addition, the pathologist usually used the first tumor section for H&E staining to access the percentage of tumor cells and then randomly selected 10 unstained sections for genomic DNA purification. The percentage of tumor cells in the H&E stained section may not reflect the tumor content in the tissue sections. Next, an algorithm named InfiniumPurify (Qin et al., Genes & Diseases, 2018) was applied to estimate the percentage of tumor cells using the RRBS data. The result indicated that the reviewer was correct. Therefore, we updated the result in Figure 3g and revised the conclusion as follows:

Not surprisingly, tumor cell content was positively correlated with prediction accuracy. (Line 344-345).

We also performed a downsampling analysis to answer the reviewer's second question. Here is what we added to the revised manuscript:

To investigate the relationship between sequencing depth and prediction accuracy in low and high tumor content groups, we divided FFPE-RRBS samples into low tumor content group (tumor content ≤ 0.5) and high tumor content group (tumor content > 0.5). Downsampling analysis of either 49 low or 20 high tumor content FFPE-RRBS libraries with 24 M reads or more indicated that increasing sequencing data could also improve the prediction accuracy (Supplementary Fig. 3). (Lines 347-352).

4. The test cohort is small (33 patients), poorly representing spectrum of cancers in CUP. The conclusions would be substantially stronger if that number is doubled at least. TCGA data is less relevant in this context as the paper is focused on the value of BELIVE in detecting primary cancer sites in CUP.

We agree with the reviewer and believe that more CUP patients need to be enrolled in this study to validate the predictive accuracy. Fortunately, 35 additional CUP patients were

included in the study. The primary tumor sites of these patients were assessed using our model and compared with the clinicopathological conclusions. The overall prediction accuracies are improved: 76-88% (old) vs. 81-93% (Table 2, new). We have updated the result in Table 2 and changed the relevant numbers in our revised manuscript accordingly.

Reviewer #3 (Remarks to the Author): expertise in machine learning using methylation data

The authors developed a novel genome-wide methylation assay FFPE-RRBS that is particularly for degraded DNA of FFPE tissues. The multi-evaluation parameters showed that the optimized FFPE-RRBS libraries were reliable for methylation profiling and provided a more profound and deeper coverage for CpGs in CGIs. They constructed 28 classifiers for predicting the origin site by combining four methylation measure methods and seven machine learning approaches. Ultimately, the optimal classifier of the mean methylation beta values-based linear support vector achieves the overall accuracies in the independent validation set of 215 metastatic cancers and successfully identified the origin site for ~76-88% of 33 CUP patients. Overall, the integration analysis is reliable. However, several main concerns are further described below:

We thank the reviewer for carefully reading the manuscript and providing constructive criticism. The kind comments encourage our team to improve the quality of this study continuously, and the excellent questions help us to address several important issues. We believe that including results from additional experiments suggested by the reviewer has significantly improved the manuscript's clarity. Our point-by-point response is shown below:

1. The authors developed a new DNA methylation profiling method FFPE-RRBS and demonstrated its reliability in several aspects. Although compared the sequencing depth of FF-RRBS and FFPE-RRBS on CGI, what is the specific performance of FF-RRBS and FFPE-RRBS in terms of CpG methylation site detection rate? What is the performance of FFPE-RRBS reproducible?

We appreciate that the reviewer raised questions to address the reliability of the FFPE-RRBS assay. In response to these questions, we have undertaken additional testing and analysis. The results have been incorporated into the new version of our manuscript, as described in the following paragraph.

However, we did observe that the FFPE-RRBS libraries had a narrow size distribution, and the mean insert library sizes were smaller compared to the FF-RRBS libraries (FF: 118 bp; FFPE: 82 bp, $P = 1 \times 10^{-10}$, **Fig. 2a and Supplementary Table 1**), consistent with an early report [59]. *Interestingly, when randomly sampling 800 M of sequencing reads, each of the FF-RRBS libraries (n=19) detected on average approximately 4.24% of CpGs at $\geq 5x$; in contrast, each of the paired FFPE-RRBS (n=19) covered about 3.50% of CpGs at $\geq 5x$ ($p = 3.92 \times 10^{-3}$, Supplementary Fig. 1a). (Lines 285-288).*

We lastly investigated the reproducibility of the FFPE-RRBS assay by selecting 12 samples with

*a sufficient amount of genomic DNA from 4 cancer types, including lung, breast, liver, and colorectal cancer, and generating triplicate libraries for each sample. As shown in **Supplementary Fig. 2**, the triplicate libraries exhibited a good correlation with the PCC values ranging from 0.84 to 0.99, indicating that our FFPE-RRBS protocol is highly reproducible. (Lines 314-319).*

2. In the feature selection part of the manuscript, one of the selection criteria is that the methylation level of a selected CGI in one cancer should show a significant difference with that of other cancers. How did they compare each of the rest cancer types? This may need to be further elaborated in the methods.

We apologize for the unclear description. Following the reviewer's suggestion, we have reworded the procedure and provided the link where we deposited our code.

Noteworthy, the CGIs should meet the following conditions: (1) The methylation level of a selected CGI in one cancer type should show a significant difference compared to other cancer types ($FDR \leq 0.01$). For this purpose, we calculated the methylation level of each CGI in one cancer type and compared it with that in the remaining nine cancer types and selected only the cancer type-specific CGI, regardless of whether it is hypomethylated or hypermethylated. (2) The selected CGI should also have a significantly different methylation level ($FDR \leq 0.01$) compared to that in the control group consisting of 57 tumor-adjacent normal tissues. The code for the selection of CGIs is available at <https://github.com/heshutao0420/cup>. (Lines 199-207).

3. In the manuscript, the authors constructed models based on the training dataset of primary FF-RRBS tissues and validated them on the FFPE-RRBS dataset of metastatic cancers. Why the authors did not initially choose to sequence and construct models based on 258 metastatic and paired primary samples? After all, this strategy is more in line with real-world molecular signatures of primary cancers and metastatic cancers. In addition, whether the methylation profile of metastatic cancer is more similar to the origin site or primary cancer occurred on the metastatic site?

We could not agree more with the reviewer regarding using paired metastatic and primary samples to generate the CUP prediction model. Unfortunately, it is extremely rare to have paired samples from the same patients in the clinic, especially considering that we need a certain number of samples for each cancer type to generate a prediction algorithm.

To investigate whether the methylation profile of metastatic cancer resembles the site of origin or primary cancer at the metastatic site, we performed a systemic comparison of the correlation between the methylation profiles of metastatic cancer tissues with those of the initial tumor tissues or the primary tumors at the metastatic sites. The conclusion has been incorporated into the text. We wrote:

In addition, we compared the methylation profiles of metastatic tumor tissues (25 liver, 8 lung, and 3 stomach tissues; Table 1) with those of the initial tumor sites and the primary tumor

occurring at the metastatic sites. The results showed that the correlation between the metastatic tissues and the initial tumor tissues was significantly higher, regardless of how the methylation alterations were calculated (Supplementary Fig. 1b). (Lines 307-312).

4. In the background, the authors mentioned the published classifier EPICUP which was established based on DNA methylation microarray data. The comparison between EIPCUP and BELIVE is necessary to be made in the performance evaluation section.

We like the reviewer's suggestion and have contacted the corresponding author, Prof Manel Esteller of the Bellvitge Biomedical Research Institute, twice to request the original data. However, we have not received a reply, even though we have cc'd the email to the editorial office of Lancet Oncology. We are therefore afraid that we will not be able to do the side-by-side comparison for the two algorithms.

-----Original Messages-----

From: "谷红仓" <gu_hongcang@cmpt.ac.cn>
Sent Time: 2023-03-28 16:21:58 (Tuesday)
To: mesteller@idibell.cat
Cc: editorial@lancet.com
Subject: DNAm microarray data

Dear Professor Esteller,

Re: DNAm microarray data

   New Meeting

2023-04-12 08:51:07

From: "谷红仓" <gu_hongcang@cmpt.ac.cn>

To: mesteller@idibell.cat

Cc: editorial@lancet.com

▶ Sent Successfully. Show detail To the recipient of 2, among them 2 is Successfully delivered to mail server

Dear Professor Esteller,

5. The authors have validated the classifier of BELIVE on an external cohort of TCGA Illumina 450K methylation arrays. How about is the performance of this method for the WGBS data?

This is an excellent suggestion. We thank the reviewer for suggesting to use WGBS data set to evaluate the classifier of BELIVE. WGBS has been considering the gold standard for genome-wide DNA methylation profiling. However, the downside of this method is that it is not cost-effective, and most sequencing reads do not contain CpGs (Ziller et al., Nature, 2013). Nevertheless, we selected 10 WGBS libraries generated from 10 metastatic tumor tissues, including breast (n=2), lung (n=1), thyroid (n=1), colon and rectum (n=2), ovary (n=2), esophagus (n=1), and stomach (n=1). We sequenced those libraries using a Nova seq6000 and got approximately 90G data and performed the evaluation accordingly. The

result has been added to the revised manuscript, as shown in the following paragraph.

Additionally, WGBS has been considered the gold standard for genome-wide methylation profiling and can cover most CGIs. We generated WGBS libraries using 10 metastatic tumor tissues, including breast (n=2), lung (n=1), thyroid (n=1), colon and rectum (n=2), ovary (n=2), esophagus (n=1), and stomach (n=1) and then performed the correlation analysis by comparing the beta value of CGIs derived from the WGBS and corresponding RRBS libraries. The results showed that all 10 paired libraries had a good correlation with PCC values ranging from 0.87 to 0.96 (Supplementary Fig. 4). Most importantly, BELIVE correctly predicted the primary sites using the WGBS dataset (Supplementary Table 6). (Lines 354-361).

6. The authors need to provide a method with sufficient detail to allow reproduction of the analysis, including any parameters related to the classifier construction for each machine learning method, such as the number of trees, the number of features evaluated each time, and the feature selection criteria (such as Gini impurity, entropy, or something else?)

We appreciate the reviewer's suggestion and believe the detailed protocol is necessary for researchers interested in our assay and classifier. The step-by-step data processing procedure has been included in the revised version of our manuscript as follows:

Specifically, we used a Bayesian optimization approach to select the hyperparameters and then divided the 498 FF-RRBS libraries into training and validation datasets according to the ratio of 4:1. To balance the number of RRBS libraries in the training and validation sets, the SMOTE function from the imblearn Python library was employed. For the training set, we used the hyperopt Python library to select the following hyperparameters: (1) Adaboost: the maximum number of estimators at which boosting stops, the learning rate; (2) KNN: the number of neighbors, the algorithm used to compute the nearest neighbors, the weight function used in the prediction; (3) LGR: the penalty, the inverse of the regularisation strength, the algorithm used in the optimization problem; (4) LinearSVC: norm used in penalization, loss function, tolerance for stopping criteria, regularisation parameter, multi-class strategy, the maximum number of iterations to run; (5) NB: additive (Laplace/Lidstone) smoothing parameter; (6) RF: maximum depth of the tree, number of features to consider when searching for the best split, number of trees in the forest, feature selection criteria; (7) SVM: regularisation parameter, kernel type to be used in the algorithm, kernel coefficient. After optimizing the hyperparameters, the prediction probability of each model was calibrated using the CalibratedClassifierCV function from the Scikit-learn package on the corresponding validation set. The codes are available from <https://github.com/heshutao0420/cup>. (Lines 212-229).

7. Across the cancer types, the performance of the machine learning model varies substantially. For cancers with relatively poor predictive performance, whether are the tumor sample size and cancer subtype heterogeneity the potential influencing factors? The authors should perform the evaluation for the potential factors?

We thank the reviewer for the constructive questions. We strongly agree with the reviewer that both sample size and pathological subtype are undoubtedly critical and may affect the predictive accuracy of our model.

To address the sample size issue, we performed additional tests. First, we performed a downsampling analysis using 55 primary colorectal cancer samples. The results showed that a larger sample size could improve the recall of our model to some extent (Figure a, below). We then used 100 primary colorectal tumors from the TCGA dataset and performed a similar analysis. The same conclusion can be drawn (Fig. b). Interestingly, 50 samples for model training give an accuracy of 0.95, suggesting that the samples we used for training in this project are sufficient.

Fig. Recall for colorectal cancer (a), and prediction accuracy of BELIVE for different bins of sample numbers.

It is generally accepted that each tumor type has a characteristic DNA methylation pattern. In addition, there is growing evidence that each tumor type may also contain multiple DNA methylation subgroups (Witte et al., Genome Medicine 2014). Consequently, it is foreseeable that prediction accuracy will be negatively affected if the tumor samples in the training set do not contain all DNA methylation subgroups. Unfortunately, this is almost unavoidable in most clinical studies due to a lack of patients. This is likely to be the case for the low prediction accuracy of head and neck cancer in our study, as patients from several pathological subgroups, which may directly or indirectly correlate with DNA methylation subgroups, were not included in our study. We thank the reviewer for pointing out this possibility and have included this point in our revised manuscript.

REVIEWERS' COMMENTS

Reviewer #1 (Remarks to the Author):

All my concerns have been adequately addressed. The revised version of the manuscript has been significantly improved and I suggest to proceed with its acceptance and publication. Congratulations on this great work.

Reviewer #2 (Remarks to the Author):

I am happy how the criticisms were addressed. The manuscript is suitable for publication.

Reviewer #3 (Remarks to the Author):

This a substantially improved version of the manuscript. The authors have addressed all my previous comments.

REVIEWER COMMENTS

Reviewer #1 (Remarks to the Author): clinical expertise in cancer of unknown primary

In this article, the authors present a novel method specifically for generating RRBS libraries using FFPE samples and develop machine learning-based classifiers for predicting the primary site of metastatic cancers. The manuscript is straightforward, well written, and concise and has clear results. Definitely deserves to be published and is a valuable contribution to the "Nature Communications" journal. Some comments need to be addressed before publication.

[1] "INTRODUCTION", Lines 69-70:

"Accurate identification of the primary site is the starting point for cancer diagnosis, and it is critical to guide the subsequent treatments of metastatic cancer [9, 10]."

In that regards, it is worthy to be reported that 28% of patients with CUP present one or more predictive biomarkers to immune checkpoint inhibitors (ICI), such as programmed death-ligand 1 (PD-L1) expression on $\geq 5\%$ cancer cells in 22.5% ($\geq 1\%$ in 34%) and lymphocytes in 58.7%, microsatellite instability (MSI)-high in 1.8% and tumour mutational burden (TMB) ≥ 17 mutations per megabase in 11.8%. However, these biomarkers are not yet validated in patients with CUP. Generally, CUP patients with TMB > 10 mutations per megabase have a trend for better outcomes when treated with ICI.

Recommended reference: Rassy E, et al. Genomic correlates of response and resistance to immune checkpoint inhibitors in carcinomas of unknown primary. *Eur J Clin Invest.* 2021;51(9):e13583.

[2] "INTRODUCTION", Lines 70-73:

"Multiple retrospective studies indicate that CUP patients who receive site-specific chemotherapy have improved overall survival (OS) compared with patients treated with empirical chemotherapy [6, 11-13]."

At that point, the authors are highly encouraged to report that there are significant deficiencies in the currently available studies comparing site-specific therapy and empiric chemotherapy. These deficiencies include patient accrual problems (oversampling treatment-resistant tumor types and long recruitment), study design limitations (observational and problematic trials), heterogeneity among the CUP classifiers (epigenetic vs. Transcriptomic profiling), and incomparable therapies. The assessment of recently published CUP literature allows to recommend two comprehensive clinical trial designs, a visionary and a pragmatic approach. Both are amenable to implementing the latest diagnostics and therapeutic advances to improve the quality of CUP research and the prognosis of many patients.

Recommended reference: Rassy E, et al. Systematic review of the CUP trials characteristics and perspectives for next-generation studies. *Cancer Treat Rev.* 2022;107:102407.

[3] "DISCUSSION", Lines 371-372:

"The treatment and potential outcome of metastatic cancer largely depend on the primary site [7, 39]."

The authors are strongly recommended to mention the case of melanoma of unknown primary (MUP), which represent approximately 3% of all melanomas. Recently has been published that patients with MUP site seem to present better outcomes compared to those with stage-matched melanoma of known primary (MKP), probably due to higher immunogenicity as reflected in the immunologically mediated primary site regression. As such, MUP patients on immunotherapy probably display better outcomes when compared to the MKP site subset.

Recommended reference: Boussios S, et al. Melanoma of unknown primary: New perspectives for an old story. *Crit Rev Oncol Hematol.* 2021;158:103208.

[4] "DISCUSSION", Lines 376-378:

"In the medical community, a widely accepted concept is that at least 15-20% of CUP patients have a favorable prognosis under the primary site-guided therapy [4]."

Please, do report that the favorable risk cancer subgroup includes patients with neuroendocrine carcinomas of unknown primary, peritoneal adenocarcinomatosis of a serous papillary subtype, isolated axillary nodal metastases in females, squamous cell carcinoma involving non-supraclavicular cervical lymph nodes, single metastatic deposit from unknown primary and men

with blastic bone metastases and PSA expression. Very recently, new favorable subsets of CUP seem to emerge including colorectal, lung and renal CUP which underlies specific treatments. Recommended reference: Rassy E, et al. New rising entities in cancer of unknown primary: Is there a real therapeutic benefit? *Crit Rev Oncol Hematol*. 2020;147:102882.

[5] "DISCUSSION", Lines 378-380:

"With increasing new treatment options, such as targeted therapies and immunotherapies, the test that can accurately identify cancer primary site will benefit thousands of cancer patients [10, 19, 64].".

Furthermore – from the therapeutic point of view – chromosomal instability (CIN) is not a frequent phenomenon in CUP, which may favour immune checkpoint inhibitors (ICI) among patients with CUP. Conversely, these patients present individual gene alterations implicated in immune-evasion and resistance to ICI. Further clinical investigations are needed to provide more information regarding the interplay between CIN, point mutations and the immune system, allowing a better understanding of ICI use in patients with CUP and potentially improving their efficacy.

Recommended reference: Chebly A, et al. Chromosomal instability in cancers of unknown primary. *Eur J Cancer*. 2022;172:323-325.

Reviewer #2 (Remarks to the Author): technical expertise in methylation sequencing

Identification of primary tumour site after detection of metastasis can be challenging. These cases are defined as cancers of unknown primary (CUP). CUP complicates treatments, which are often tailored to the origin of tumours. In this manuscript Zhang et al aimed to develop DNA methylation-based classifier, which would identify primary site of cancer. First, authors adapted reduced representation bisulfite sequencing method (RRBS) to be applied to the formalin fixed paraffin-embedded tissues (FFPE). DNA methylation reference data was collected from the cohort of 498 fresh frozen primary tumour samples and validation cohort was extracted from 215 FFPE samples. Beta value-based linear support vector classifier (BELIVE) demonstrated the best performance. Finally, authors employ the developed classifier to detect primary tumour sites on 33 CUP patients with 88% accuracy.

While the manuscript presents potentially important advancements in identifying cancer origin in CUP patients, the manuscript lacks details, which would enable the study to be reproduced and broadly used by a biomedical community. Specific criticisms are as follows:

1. In the abstract authors claim to present a novel DNA methylation detection method, however there is not enough details presented, which would enable evaluation of the novelty of the method. In materials and methods there is a superficial description of RRBS with reference to paper by Boyle P et al (2012), which describes much longer version of the protocol. In the main text authors mention the dephosphorylation before MspI digestion and the choice of the buffer. Authors must present all the details (in materials and methods part), including enzyme manufacturers, concentrations, incubation times etc such that the community would be able to benefit from the methodological discoveries.
2. The RRBS data must be made available in publicly accessible data repository such a short read archive (and accession number indicated).
3. In Fig 3g the first bar should be labelled 0.1 to 0.2 based on what is indicated in text (>10% tumour content). I find it very surprising that the accuracy does not correlate positively with tumour content. Is tumour content reflected in sequencing data as it is in the histological samples? What is relationship between sequencing depth (read numbers) and accuracy in low and high tumour contents groups?
4. The test cohort is small (33 patients), poorly representing spectrum of cancers in CUP. The conclusions would be substantially stronger if that number is doubled at least. TCGA data is less relevant in this context as the paper is focused on the value of BELIVE in detecting primary cancer sites in CUP.

Reviewer #3 (Remarks to the Author): expertise in machine learning using methylation data

The authors developed a novel genome-wide methylation assay FFPE-RRBS that is particularly for degraded DNA of FFPE tissues. The multi-evaluation parameters showed that the optimized FFPE-RRBS libraries were reliable for methylation profiling and provided a more profound and deeper coverage for CpGs in CGIs. They constructed 28 classifiers for predicting the origin site by combining four methylation measure methods and seven machine learning approaches. Ultimately, the optimal classifier of the mean methylation beta values-based linear support vector achieves the overall accuracies in the independent validation set of 215 metastatic cancers and successfully identified the origin site for ~76-88% of 33 CUP patients. Overall, the integration analysis is reliable. However, several main concerns are further described below:

1. The authors developed a new DNA methylation profiling method FFPE-RRBS and demonstrated its reliability in several aspects. Although compared the sequencing depth of FF-RRBS and FFPE-RRBS on CGI, what is the specific performance of FF-RRBS and FFPE-RRBS in terms of CpG methylation site detection rate? What is the performance of FFPE-RRBS reproducible?
2. In the feature selection part of the manuscript, one of the selection criteria is that the methylation level of a selected CGI in one cancer should show a significant difference with that of other cancers. How did they compare each of the rest cancer types? This may need to be further elaborated in the methods.
3. In the manuscript, the authors constructed models based on the training dataset of primary FF-RRBS tissues and validated them on the FFPE-RRBS dataset of metastatic cancers. Why the authors did not initially choose to sequence and construct models based on 258 metastatic and paired primary samples? After all, this strategy is more in line with real-world molecular signatures of primary cancers and metastatic cancers. In addition, whether the methylation profile of metastatic cancer is more similar to the origin site or primary cancer occurred on the metastatic site?
4. In the background, the authors mentioned the published classifier EPICUP which was established based on DNA methylation microarray data. The comparison between EIPCUP and BELIVE is necessary to be made in the performance evaluation section.
5. The authors have validated the classifier of BELIVE on an external cohort of TCGA Illumina 450K methylation arrays. How about is the performance of this method for the WGBS data?
6. The authors need to provide a method with sufficient detail to allow reproduction of the analysis, including any parameters related to the classifier construction for each machine learning method, such as the number of trees, the number of features evaluated each time, and the feature selection criteria (such as Gini impurity, entropy, or something else?)
7. Across the cancer types, the performance of the machine learning model varies substantially. For cancers with relatively poor predictive performance, whether are the tumor sample size and cancer subtype heterogeneity the potential influencing factors? The authors should perform the evaluation for the potential factors?